# Altered Postcentral Connectivity after Sleep Deprivation Correlates to Impaired Risk Perception: A Resting-State Functional Magnetic Resonance Imaging Study

**DOI:** 10.3390/brainsci13030514

**Published:** 2023-03-20

**Authors:** Jie Chen, Xinxin Gong, Letong Wang, Mengmeng Xu, Xiao Zhong, Ziyi Peng, Tao Song, Lin Xu, Jie Lian, Yongcong Shao, Xiechuan Weng

**Affiliations:** 1School of Psychology, Beijing Sport University, Beijing 100084, China; 2School of Biological Science and Medical Engineering, Beihang University, Beijing 100083, China; 3Department of Neuroscience, Beijing Institute of Basic Medical Sciences, Beijing 100850, China

**Keywords:** regional homogeneity, postcentral gyrus, voxel-wise functional connectivity, sleep deprivation, risk perception

## Abstract

Background: Previous studies revealed that sleep deprivation (SD) impairs risk perception and leads to poor decision-making efficiency. However, how risk perception is related to brain regions’ communication after SD has not been elucidated. In this study, we investigated the neuropsychological mechanisms of SD-impaired risk perception. Methods: Nineteen healthy male adults were recruited and underwent resting-state functional magnetic resonance imaging during a state of rested wakefulness and after nearly 36 h of total SD. They then completed the balloon analog risk task, which was used to measure the risk perception ability of risky decision-making. Regional homogeneity (ReHo) and voxel-wise functional connectivity were used to investigate neurobiological changes caused by SD. Correlation analysis was used to investigate the relationship between changes in ReHo, function, and risk perception. Results: At the behavioral level, risk perception decreased after 36 h of SD. At the neural level, SD induced a significant increase in ReHo in the right postcentral gyrus and was positively correlated with risk perception changes. The functional connectivity between the right postcentral gyrus, left medial temporal gyrus, and right inferior temporal gyrus was enhanced. Critically, increased right postcentral gyrus and right inferior temporal gyrus connectivity positively correlated with changes in risk perception. Conclusions: SD impairs the risk perception associated with altered postcentral connectivity. The brain requires more energy to process and integrate sensory and perceptual information after SD, which may be one possible reason for decreased risk perception ability after SD.

## 1. Introduction

With the high pressure and fast pace of life in modern society, a lack of sufficient sleep has become a common problem. Sleep loss can lead to various brain dysfunctions, poor behavioral performance, and other serious consequences [1]. Sleep deprivation (SD), commonly defined as staying awake for more than 24 h, is an experimental model of temporary impairment that sheds light on insufficient sleep, which affects brain function and behavioral performance [2]. Impairment of cognitive function by SD presents as a step-down, with two trough periods and plateaus. The first trough period occurred within 24–48 h of SD, which caused a 20–30% decline in cognitive function. In addition, 48–72 h of SD is the second trough period, with a 40–50% decline in cognitive function [3]. Previous studies have shown that SD can strongly impair human cognitive function, including attentional lapses [4], delayed sensations [5], impaired working memory [6], uncontrollable emotions [7], and confused thinking [8]. However, few studies have investigated the neural mechanisms underlying advanced cognitive functions, particularly risky decision-making. Decision-making is a jewel in the crown of consciousness, in which cognitive processes are very complex, including sensory perception [9], anticipation [10], feature recognition [11], working memory [12], emotion [7], intuition [13], insight [14], and creative thinking [15], which are a series of logical and illogical components. Risky decision-making is an uncertain decision-making process that occurs in the case of uncertain probability and results. The most widely used paradigm for risky decision-making is the balloon analog risk task (BART), which uses a computer to simulate risky decision-making in the real world to judge and predict individuals’ risky decision-making preferences [16]. Lack of sleep may have a negative impact on risky decision-making. However, results on how SD affects risky decision-making remain inconsistent [17,18,19]. Some studies have shown that after SD, participants are more risk-seeking, while other studies have reached the opposite conclusion [20,21]. This suggests that different variables cause SD to affect risky decision-making in different ways, leading to varying results.

Risk perception refers to individuals’ subjective feelings and cognition of risks in decision-making, that is, the extent to which decision-makers regard uncertainty as a threat or opportunity [22]. Risk perception includes the probability estimate of the uncertainty and controllability of the situation and the estimate of confidence; the key is the estimate of the relative risk level of the potential outcome [23]. A systematic review found that risk perception, risk tolerance, and risky decision-making strategies affect decision-making behavior after SD [24]. SD impairs risk perception and increases risk as a threat, resulting in poor decision-making efficiency [25]. Lin et al. (2015) found that compared with healthy people, the activation of the inferior frontal gyrus and anterior central gyrus decreased in patients with online gaming disorder when comparing probabilistic options [26]. Zhang et al. (2014) explored the developmental cognitive neural mechanisms of risky decision-making in adolescents and concluded that the parietal lobule, medial parietal lobe, lateral prefrontal lobe, insula, and amygdala were correlated with risk perception and probabilistic processing [27]. No neuron operates in isolation. The way that brain regions involved in risk perception communicate with each other after SD has not as yet been elucidated. Therefore, we investigated the neuropsychological mechanisms of SD-impaired risk perception. By gaining a better understanding of how decision-makers evaluate risks and benefits under different scenarios, we can improve our ability to make risky decisions with fewer negative consequences.

Functional magnetic resonance imaging (fMRI) is a high-spatial-resolution imaging technique based on blood oxygenation level dependence that records magnetic induction changes in the brain without damage. It is the most widely used method in cognitive neuroscience [28]. The ventromedial prefrontal lobe is considered the core brain region for risky decision-making and is mainly responsible for storing and marking the value of future outcomes. SD leads to ventromedial prefrontal activation, which increases an individual’s expectation of a reward [29]. In addition to the prefrontal regions, the insula and temporal lobes also play important roles in risky decision-making. In the context of active risky decision-making, activation of the left insula and right superior parietal lobe was moderately correlated with risky behavior [30], and reduced insula activation led to reduced sensitivity to loss [29]. In the case of brain injury, it has been found that the temporal lobe is involved in the risky decision-making process and is associated with feedback learning [30]. The superior temporal gyrus, which integrates prior behaviors and successful outcomes, affects individual decision-making strategies, while the middle temporal gyrus is correlated with sensory conflict control of feedback [31]. Therefore, we speculate that the postcentral gyrus and prefrontal and temporal lobes also play important roles in risk perception.

Regional homogeneity (ReHo) is an indicator of resting brain activity used to measure local functional connectivity. Focusing on the specialization of brain functions can help us to understand more complex and advanced cognitive brain functions [32]. Dai et al. (2012) used the regional homogeneity method and found that ReHo changes in various brain regions after SD were correlated with brain dysfunction, with emotional regulation recruitment being the most obvious. In addition to processing auditory information, the temporal lobe is involved in memory and emotion [33]. SD can lead to emotional changes that affect an individual’s perceptions of decision-making opportunities and threats. Voxel-wise functional connectivity is a whole-brain functional connectivity method based on voxels that can help better integrate brain functions. The inferior temporal gyrus is primarily involved in processing visual information. As a dorsal recruitment of vision, the functional connectivity between the inferior temporal gyrus and the postcentral gyrus perceives spatial information [34]. SD impairs the processing of visual information and thus affects the early processing of decision-making. Studies have shown that the brain is not a fully functionally differentiated system, different brain regions interact with information all of the time, and people need the collaborative participation of multiple brain regions to complete any task. Risky decision-making is a high-level cognitive function of the brain that requires collaborative cooperation between multiple brain regions [35]. Therefore, a functional brain network connectivity analysis is more suitable for exploring the neuropsychological mechanisms of SD-impaired risk perception.

In this study, we investigated the neuropsychological mechanisms of SD-impaired risk perception. First, we calculated the changes in the regional homogeneity of the brain after SD to identify brain regions with noteworthy changes. The brain regions with a significant correlation between the changes in ReHo and the changes in risk perception were used as seed regions for subsequent functional connectivity analysis. We hypothesized that the ReHo of brain regions related to probabilistic processing and risk perception, such as the parietal lobule, medial parietal lobe, lateral prefrontal lobe, inferior frontal gyrus, precentral gyrus, insula, and amygdala [27], would change substantially and would be significantly correlated with risk perception changes. Brain regions with significant changes in ReHo after SD were used as seed regions for the voxel-wise functional connectivity analysis. Finally, we correlated the significant changes in functional connectivity with changes in risk perception after SD. We hypothesized that SD impairs risk perception, and that the postcentral gyrus plays an important role in risk perception.

## 2. Materials and Methods

### 2.1. Participants

Nineteen healthy male adults were recruited (mean ± SD, age 21.79 ± 2.37 years, all right-handed) from Beijing Sport University. Participants’ IQ was above the population average (IQ > 110). Participants’ scores on the Pittsburgh Sleep Quality Index (PSQI < 5) indicated good sleep quality. Inclusion criteria were: (1) A healthy body with normal naked eye vision or corrected vision; (2) No history of major diseases, no history of brain trauma, no metal implants in the body, no neurological, mental, and sleep disorders; (3) No habit of smoking, drinking, or other drug addiction; (4) No trans-meridian travel, shift work or irregular sleep/wake routines in the 60 days prior to the in-laboratory experiment; (5) No caffeine or medication within 48 h before each scan; and (6) The morning and night questionnaire showed normal sleep pattern, and the Pittsburgh Sleep Quality Index score was below 5, indicating good sleep. The Ethics Committee of Beihang University (Beijing, China) approved this study. All participants provided informed consent. At the end of the experiment, the participants were paid a specific monetary sum related to their participation.

### 2.2. Experimental Protocol

In this study, we used a within-subject, repeated-measures, counterbalanced experimental design to investigate the neuropsychological mechanisms of SD-impaired risk perception. All participants underwent two scans, one during rested wakefulness (RW) and one after nearly 36 h of total SD at least one week apart. We used a balanced approach to avoid confounding factors in the study design, with half of the participants undergoing an RW scan followed by an SD scan. Before the experiment, participants were asked to participate in a semi-structured interview as a screening session to ensure that they met all inclusion criteria. Participants who met the requirements were brought to the laboratory three times. For the first time, they were required to arrive at the lab by 4 p.m., complete an informed consent form and a demographic questionnaire, and learn about lab procedures and precautions. For the second time, they arrived at the laboratory at 8 a.m. During RW, the participants underwent MRI and a BART. Under SD, the participants were required to stay awake for 36 h from 8 a.m. on the first day of the formal experiment to 8 p.m. on the second day. During SD, the participants were only allowed to engage in light exercises, such as writing or reading, while keeping their eyes open. Participants underwent MRI scan at 8 p.m. after 36 h of SD. During the MRI scan, the participants were asked to keep their heads stable and their eyes focused on the “+”. After the scan, the participants were asked to actively report whether they fell asleep during the scan and finally complete the BART. The experimental protocol is illustrated in Figure 1.

### 2.3. The Balloon Analog Risk Task (BART)

In this study, in the BART, we only considered the risky decision-making behavior tendency of participants in the active mode (participants could inflate the balloon by pressing a button, and there was a certain probability that it would explode). During the experiment, a virtual blue balloon appeared at the center of the screen. Participants could choose to inflate the balloon by pressing different buttons (each time they inflated the balloon, they would receive a monetary reward of CNY 0.2, and all monetary rewards obtained in the game would be converted into the same amount of cash to be returned to the participants after the experiment) or accept the temporary monetary rewards obtained by the current accumulation. The number of times each balloon exploded was random (following a uniform distribution between 1 and 31). Each time the participants completed a selection, they had a random rest period of 1–3 s (subject to a uniform distribution), and then made the next selection. Each balloon had a trial period from the beginning of inflation to the explosion, or the participant chose to accept the monetary reward. There was a random rest period of 2–6 s between trials. Forty trials were conducted. Figure 2 shows the balloon analog risk task.

In the BART, the main behavioral metrics are the number of explosions, BART value, and total earnings. The number of explosions was used to measure the failure performance of decision-making, total income was used to measure the success performance of decision-making, and the BART value was used to measure the tendency to take risks in the process of risky decision-making. The BART value refers to the average number of inflations of an unexploded balloon. The higher the BART value, the more inclined the participant is to take risks, which is the most sensitive indicator of BART. Based on the unexploded balloon, we introduced the risk value, which refers to the number of inflations of the unexploded balloon divided by the predetermined number of explosion inflations, and then averaged it to obtain the average possibility of the distance between each unexploded balloon and the explosion. Risk values were used to measure participants’ risk perception. The higher the risk value, the closer the distance from the explosion, and the more sensitive the risk perception; conversely, the farther the distance, the less sensitive the risk perception, which was highly correlated with the number of explosions, BART value, and total earnings, with correlation coefficients of 0.79, 0.95, and 0.75, respectively, and *p* < 0.001. In conclusion, risk values can be used as reliable indicators to measure risk perception in participants’ risky decision-making processes.

### 2.4. Magnetic Resonance Imaging (MRI) Data Acquisition

All neuroimaging data were collected under both RW and SD conditions using a 3-T SOMATOM Magnetom Skyra (Siemens AG, Munich, Germany) with a Birdcage RF Head Coil at the Eighth Medical Center of the People’s Liberation Army. T2* weighted echo planar imaging sequence was used for standard resting state functional data (repeat time (TR) = 2000 ms, echo time (TE) = 30 ms, flip angle = 90°, field of view (FOV) = 256 × 256 mm^2^, matrix size = 64 × 64 mm^2^, slice = 35, slice thickness = 3 mm, 2 × 2 × 2 mm^3^), 240 volumes were collected, and the duration was 480 s. High resolution T1-weighted structural imaging (TR = 2200 ms, TE = 2.45 ms, flip angle = 8°, field of view (FOV) = 256 × 256 mm^2^, slice thickness = 1 mm) was obtained by magnetic preprocessing rapid gradient echo sequence in 1 × 1 × 1 mm^3^, 176 slices. Functional images were scanned in the axial direction, whereas T1 structural images were scanned in the sagittal direction. The participants lay in the scanner with their heads comfortably held in place using foam sponge pads to reduce head movement and earplugs to mute the noise of the scanner. During the resting-state scan, all participants were asked to stare at the “+” position with their eyes open, lie as still as possible, and not think about anything. To ensure that the participants did not fall asleep during the scan, we monitored them with a video camera and used a microphone to remind them to remain awake if necessary. At the end of each trial, the participants were asked if they had stayed awake, and all reported that they did.

### 2.5. Resting-State Functional Magnetic Resonance Imaging (fMRI) Preprocessing

The MATLAB R2021b platform (MathWorks, Inc., Natick, MA, USA), SPM12 (University College London), and RESTplus toolbox 1.27 [36] were used to preprocess the resting-state functional images. The preprocessing process included the following steps: (1) The first ten time points were removed, avoiding the instability of magnetic field signals caused by the MRI scanner and ensuring that the participants were in a resting stable state. (2) Slice timing was measured to avoid image differences caused by different sampling times. There were 35 slices in this study, and the middle slice was selected as the reference slice (slice 35) to simultaneously correct the acquisition time of all slices to the same time. (3) Head motion correction was implemented to avoid incomplete alignment at time points caused by the participant’s head displacement, thus affecting the subsequent statistical analysis. The whole-brain images at all time points were aligned with the whole-brain images at the first time point using a rigid body transformation, and six head motion parameters were obtained, including three translational and three rotational parameters. (4) Spatial standardization was applied to avoid difficulties in the horizontal analysis of subsequent groups caused by size differences in the participants’ heads. A standard spatial template from the Montreal Neurological Institute (MNI) was used for indirect registration. We first aligned the T1-weighted structural image of high spatial resolution with the resting-state functional image to obtain a transformation relationship. Next, we segmented the aligned T1-weighted structural image into different brain tissue maps, including gray matter, white matter, and cerebrospinal fluid, and aligned the structural image with the standard MNI template by affine transformation to obtain another transformation relationship. Finally, a total change relation was calculated from these two relations and applied to the resting-state functional image to achieve spatial standardization of the resting state functional image. (5) The detrend was removed to avoid a constant rise in the BOLD signal caused by the high machine temperature of the MRI scanner. (6) Regression covariables were observed to avoid noise confounding, improve the signal-to-noise ratio, and improve statistical efficacy. We adopted the Fisher 24 head motion parameters and added the first derivative and its square to the original six head motion parameters to minimize the noise of head motion. We implemented regression of cerebrospinal fluid and white matter signals to reduce the impact of noise, such as breathing and heartbeats. (7) Bandpass filtering was implemented, which avoids the influence of low- and high-frequency noise on the signal; the filtering range was 0.01–0.08 Hz.

Before further processing, we verified the quality of the preprocessed images. Three participants with head movements of >2 mm or 2° were excluded. The spatial standardization of the functional images was checked individually, and the alignment of the remaining 16 participants was as expected. Finally, all functional and structural images of 16 participants were checked for abnormalities in RW and SD, and no abnormalities were found in images of 16 participants. Therefore, a total of 16 participants were finally included in the following data analysis.

### 2.6. Regional Homogeneity (ReHo) Analysis

ReHo was calculated using the Kendall harmony coefficient to measure local functional connectivity. The cluster size corresponded to 27 voxels. It was smooth, avoided the influence of high-frequency noise, and improved statistical effectiveness. The size of the Gaussian smooth kernel was represented by the half-peak full width (half), set as 6 × 6 × 6 mm^3^ in this study. SPM12 was used to conduct a second level analysis of the smoothed normalized ReHo images. The matrix was designed as a paired-sample t-test to compare the differences in ReHo between RW and SD. To avoid the influence of air tissue, a gray matter mask was used to control the false-positive rate. Gaussian random field theory (GRF) in the RESTplus toolbox was used for multiple corrections at the uncorrected voxel level of *p* < 0.001 and at the corrected cluster level of *p* < 0.05. Finally, xjView (a viewing program for SPM) and BrainNet Viewer [37] were used to view the results. In this study, automatic anatomical labeling and Brodmann’s Interactive Atlas were used to locate the spatial structures of significant brain regions.

To further investigate the relationship between the altered ReHo brain regions and risk perception, we calculated the ReHo change by subtracting the ReHo of RW from the ReHo of SD for each participant. The change in risk perception was calculated by subtracting the value of risk in SD from that in RW. Pearson’s correlation analysis was used to calculate the relationship between changes in ReHo and risk perception.

### 2.7. Functional Connectivity Analysis

The peak coordinates of the ReHo-significant brain region were defined as the seed region, and the sphere radius was 5 mm. A voxel-wise functional connectivity analysis of the whole brain was performed. We used a paired sample t-test to compare the differences in global brain functional connectivity between SD and RW conditions. GRF was used for multiple corrections at the uncorrected voxel level, *p* < 0.001, and at the corrected cluster level, *p* < 0.05.

To further investigate the relationship between functional connectivity and risk perception changes, we first calculated changes in functional connectivity by extracting the whole brain functional connectivity matrix after SD minus the RW whole brain functional connectivity matrix. The change in risk perception was then calculated by subtracting the value of risk in SD from the value of risk in RW. Finally, Pearson’s correlation analysis was used to calculate the relationship between changes in functional connectivity and risk perception.

## 3. Results

### 3.1. Basic Statistics of Risk Decision Performance

As shown in Table 1, compared to RW, the risk perception ability of participants after SD decreased significantly (*t* = −2.19, *p* = 0.04). For RW, the mean ± standard deviation of the participants’ risk perception was 0.61 ± 0.07. Mean ± standard deviation of risk perception was 0.58 ± 0.08 after SD. Results showed that risk perception was impaired after SD.

### 3.2. Impact of Sleep Deprivation on Regional Homogeneity

As shown in Table 2, compared with RW, the right postcentral gyrus (MNI coordinates: 69, −6, 21) of the participants after SD was located in the Brodmann 42 area, and the ReHo increased significantly (t = 7.03, p-FDR < 0.05) (see Figure 3). Changes in the right postcentral gyrus ReHo were positively correlated with changes in risk perception; however, the margin was significant (r = 0.48, *p* = 0.06) (see Figure 4). These results suggest that changes in the ReHo in the right postcentral gyrus may be related to risk perception.

### 3.3. Impact of Sleep Deprivation on Functional Connectivity

As shown in Table 3, compared with RW, the right postcentral gyrus (MNI coordinates: 69, −6, 21) and left middle temporal gyrus (MNI coordinates: −51, 0, −15), which were located in the Brodmann 21 region, functional connectivity was significantly enhanced after SD (t = 7.04, p-FDR < 0.05). In the right postcentral gyrus (MNI coordinates: 69, −6, 21) and the right inferior temporal gyrus (MNI coordinates: 48, −18, −21), located in the Brodmann 21 region, functional connectivity was significantly enhanced after SD (t = 6.94, p-FDR < 0.05) (see Figure 5). Changes in the functional connectivity of the right postcentral and right inferior temporal gyri were positively correlated with changes in risk perception; however, the margins were significant (r = 0.47, *p* = 0.06) (Figure 6). These results suggest that functional connectivity between the right postcentral gyrus and right inferior temporal gyrus may be related to the compensation of risk perception.

## 4. Discussion

In this study, we investigated the neuropsychological mechanisms of SD-impaired risk perception. We calculated the changes in ReHo and voxel-wise functional connectivity during RW and SD and explored the relationship between ReHo and voxel-wise functional connectivity and risk perception changes. The results of this study showed that, at the behavioral level, the risk value of individuals decreased significantly after SD. At the neural level, regional homogeneity in the right postcentral gyrus increased significantly after SD and positively correlated with changes in risk perception. Functional connectivity between the right postcentral gyrus, the left medial temporal gyrus, and the right inferior temporal gyrus was enhanced, and changes in functional connectivity between the right postcentral gyrus and the right inferior temporal gyrus were positively correlated with changes in risk perception. These results indicate that risk perception is impaired after SD and that the right postcentral gyrus is correlated with risk perception. The enhanced connectivity between the right postcentral gyrus and right medial temporal gyrus compensates for the risk perception function, indicating that the brain needs to invest more energy in sensory and perceptual information processing and integration.

Numerous studies have shown that SD impairs cognitive function [38,39,40], which is consistent with our research results and presents a step down with two trough periods and plateaus. The first trough period occurs within 24–48 h of SD, which causes a 20–30% decline in cognitive function. In addition, 48–72 h of SD is the second trough period, with a 40–50% decline in cognitive function. Shao suggested that 24 h of SD had little impact on advanced cognitive functions, such as risky decision-making, and may not have reached the first trough of cognitive function. However, we chose 36 h of SD based on the trade-off between practical problems in the experiment and the physical and mental health problems of the participants [3]. Moreover, 36 h of SD could explain the neural mechanism of higher cognitive function impairment in the participants. Previous SD studies may have been inconsistent because of the duration of SD or degree of cognitive impairment. With prolonged deprivation, brain function is disrupted. Overactivation of the prefrontal lobe is a compensatory function, which is one of the brain’s unique functions for maintaining normal cognitive activities [41]. The compensatory mechanism was originally proposed based on Kahneman’s attentional resource allocation model, which assumes that adjusting goals and actions requires a compensatory control mechanism that dynamically allocates resources. Drummond found that task difficulty induces brain functional compensation after SD and that the parietal lobe plays an important role in cognitive function [42].

The most important finding of this study was that the right postcentral gyrus participates in the process of risky decision-making and is correlated with risk perception, which is consistent with previous research results [26,27]. After SD, the ReHo in the right postcentral gyrus, the core node of the sensorimotor network, was significantly higher than that during rested wakefulness [43]. The sensorimotor network is responsible for sensing the input of external physical stimuli, converting them into electrical signals for transmission in the brain network, forming a sensory experience, and executing the corresponding action responses [44]. Sensorimotor networks play important roles in human sensory and perceptual integration, daily life, and motor functions [45]. The postcentral gyrus is located in the anterior longitudinal gyrus of the parietal lobe between the precentral and postcentral sulci and is responsible for integrating somatosensory functions [46]. Lin found that compared with healthy people, the activation of the inferior frontal gyrus and precentral gyrus decreased in patients with online gaming disorders when comparing probabilistic options [26]. Therefore, our research results show that the right postcentral gyrus is responsible for probability processing in risk perception and that individuals constantly weigh the relative risk level of potential outcomes in the decision-making process. In RW, individuals are more likely to view risks as opportunities, while after SD, they are more likely to view risks as threats, which is a manifestation of impaired risk perception ability.

Another finding of this study was that the functional connectivity between the right postcentral gyrus and the left medial temporal gyrus was enhanced after SD compared with rested wakefulness. The postcentral gyrus plays an important role in stimulus discrimination, visual information processing, and spatial information positioning [47]. Liang found that a decline in BOLD signals in the right postcentral gyrus in patients with schizophrenia was associated with visual, auditory, and cognitive dysfunction [48]. The postcentral gyrus controls somatosensory perception and its dysfunction affects cognitive function. Dai et al. used the regional homogeneity method and found that changes in ReHo in various brain regions after SD were correlated with brain dysfunction, and changes in emotional regulation recruitment were the most obvious [33]. In addition to processing auditory information, the temporal lobe is associated with memory and emotion and is involved in risky decision-making processes [30]. The middle and inferior temporal gyri are located in regions 21 and 20 of Brodmann, respectively, and the middle temporal gyrus is a brain region with complex functions. Zhu found that the left middle frontal gyrus is mainly involved in cognitive processes, such as language recognition, understanding, and judgment [49], and is also correlated with logical reasoning [50]. Yun et al. (2017) used graph theory to show that social anxiety disorder correlates with the left middle temporal gyrus. The enhanced connectivity between the left middle temporal gyrus and insula makes the function of the social-emotional network more stable and less affected by anxiety [51] Functional disorders of the brain may occur after SD, and enhancement of the right postcentral gyrus and left middle temporal gyrus may help the brain find a balance and maintain normal primary sensory and perceptual functions to some extent. The inferior temporal gyrus is primarily involved in processing visual information. As a dorsal pathway of vision, it is functionally connected to the postcentral gyrus and perceives spatial information [47]. The functional connectivity between the right postcentral gyrus and the right inferior temporal gyrus was enhanced, and the changes in functional connectivity were positively correlated with the changes in risk value, suggesting that the brain compensated for risk perception and invested more energy in processing and integrating sensory and perceptual information [17]. SD may impair the neural circuitry underlying risk perception.

This is the first study to investigate the neuropsychological mechanisms of SD-impaired risk perception. SD reduces an individual’s risk perception ability, and more people regard risk as a threat, leading to a decrease in decision-making efficiency [25]. Our study conforms to this conclusion and provides empirical evidence that SD impairs risk perception. Studies have found that the parietal lobule, medial parietal lobe, lateral prefrontal lobe, inferior frontal gyrus, precentral gyrus, insula, and amygdala are related to probabilistic processing and risk perception [26,27], which supports our finding that changes in the functional connectivity of the postcentral gyrus are associated with impaired risk perception. Our findings deepen the relationship between SD, risk perception, and risky decision-making and provide a scientific understanding of the risky decision-making process and better decision-making behavior performance.

This study had several limitations. First, the sample size was not representative. Our sample size was small, and all of the participants were male, which affects the objectivity and universality of our research results to some extent. A previous meta-analysis showed that women were more risk averse than men [52] and showed greater risk aversion in probabilistic risk-selection tasks [53]. Therefore, sex may have affected our results to some extent, and we will provide better control for this additional variable in future studies. Second, although the participants were required to stay awake during SD and self-report whether they fell asleep after the scan, we still had no objective evidence to exclude them from falling asleep during the scan. In the future, electrophysiological techniques could be used to determine whether the participants fall asleep. Third, no control group was included. Sleep-deprivation experiments require a control group to avoid interference from additional factors, such as anticipation or learning effects. Finally, risky decision-making processes are complex. Decision-making is an advanced cognitive function that requires the collaboration of multiple brain regions and involves various cognitive processes, including risk perception, risk assessment, expected outcomes, working memory, and effective communication. The neural mechanisms underlying the decision-making process are not limited to a single technique. In addition to high spatial resolution fMRI technology, it is necessary to use high temporal resolution electroencephalography (EEG) technology to investigate dynamic decision-making brain networks by combining time-domain, frequency-domain, and time-frequency methods. Therefore, future studies should explore the neural mechanisms underlying the effect of SD on risky decision-making in terms of dynamic functional connectivity using synchronous EEG-fMRI.

## 5. Conclusions

In this study, we investigated the neuropsychological mechanisms of SD-impaired risk perception. Using the ReHo and voxel-wise functional connectivity methods, we found that at the behavioral level, the risk value of the participants decreased after SD. At the neurological level, the ReHo of the right postcentral gyrus increased after SD and was positively correlated with the change in risk value. Functional connectivity between the right postcentral gyrus, left medial temporal gyrus, and right inferior temporal gyrus was enhanced, and the change in functional connectivity between the right postcentral gyrus and right inferior temporal gyrus was positively correlated with the change in risk value. These results suggest that risk perception is impaired after SD and that this impairment is correlated with altered postcentral functional connectivity. The brain requires more energy to process and integrate sensory and perceptual information after SD, which may be one possible reason for decreased risk perception ability after SD.

## Figures and Tables

**Figure 1 brainsci-13-00514-f001:**
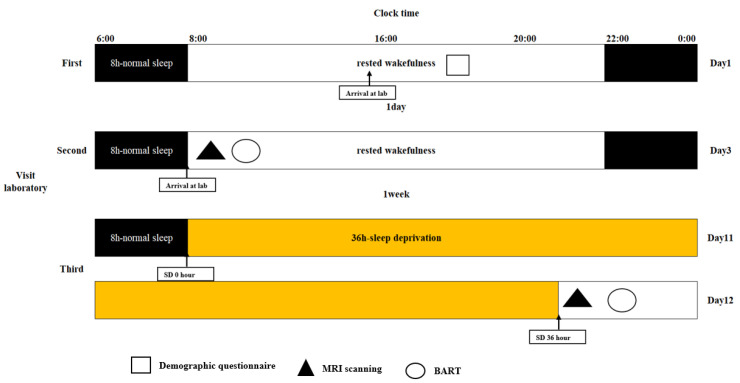
Experimental protocol. Eligible participants visited the laboratory three times. First, they arrived at the laboratory before 4 p.m., and then provided the written informed consent, received the demographic questionnaire and indicated that they understood the experimental protocol. Second, they underwent MRI scanning, after which they completed the BART during rested wakefulness. Third, they began sleep deprivation at 8 a.m., which ended at 8 p.m. on the second day. After 36 h sleep deprivation, MRI scanning was performed before the BART. The experiment lasted for 12 days. MRI, magnetic resonance imaging; BART, balloon analog risk task.

**Figure 2 brainsci-13-00514-f002:**
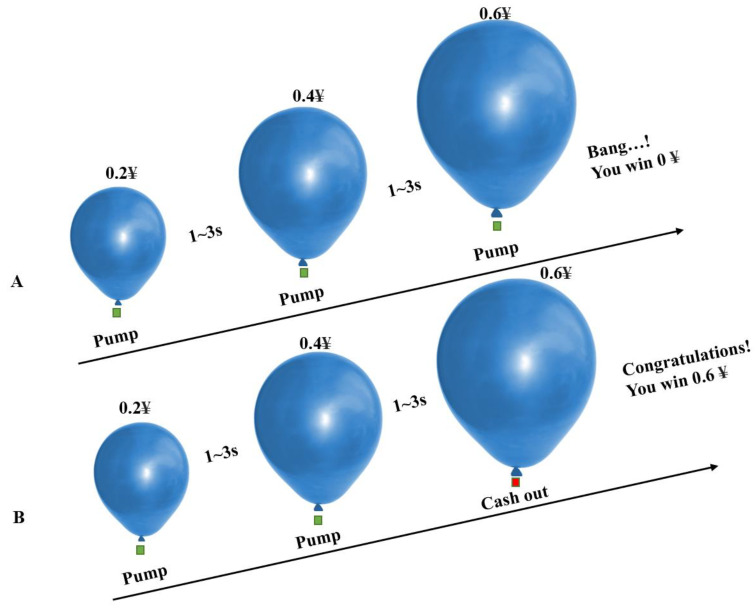
Balloon analog risk task. A exploded balloon trials. B unexploded balloon trials. A virtual uninflated blue balloon is shown in the center of the screen. Participants can press the green button to inflate balloon or red button to discontinue inflation. Participants can receive CNY 0.2 reward for each press of the green button if the balloon continues to inflate, or lose all of their reward if the balloon explodes. Each time the participants completed a selection, the game would have a rest period of 1 to 3 s; there was a rest period of 2 to 6 s between trials, and a total of 40 trials.

**Figure 3 brainsci-13-00514-f003:**
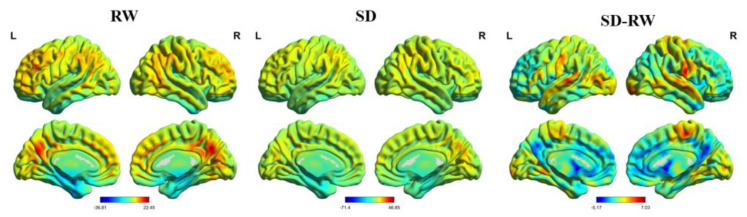
Regional homogeneity map. Compare to RW, right Postcentral regional homogeneity significant increased following SD. RW, rested wakefulness; SD, sleep deprivation.

**Figure 4 brainsci-13-00514-f004:**
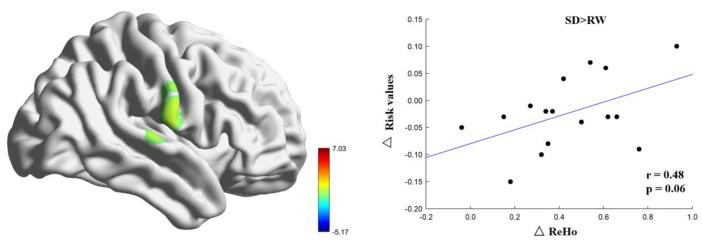
Regional homogeneity results. Regional homogeneity in the region of the right postcentral significantly increased following sleep deprivation. The corresponding brain region is illustrated at left, beside the scatter plot. Regional homogeneity (ReHo) change correlated with risk values change following sleep deprivation. SD, sleep deprivation; RW, rested wakefulness; BART, balloon analog risk task.

**Figure 5 brainsci-13-00514-f005:**
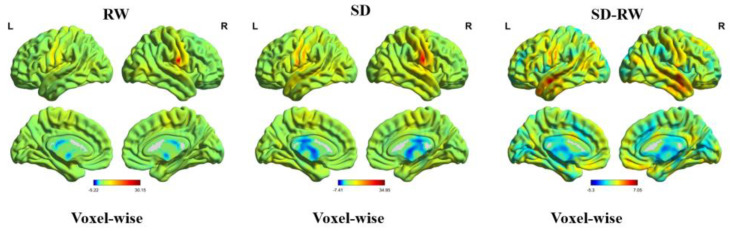
Voxel-wise functional connectivity map. Compared to RW, right postcentral and left middle temporal and right inferior temporal functional connectivity significantly increased following SD. RW, rested wakefulness; SD, sleep deprivation; R, right; L, left.

**Figure 6 brainsci-13-00514-f006:**
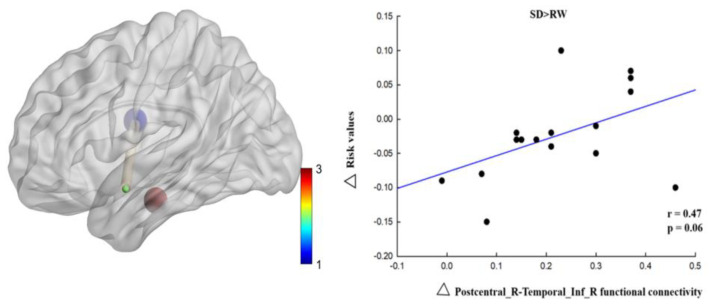
Functional connectivity changes correlated with risk value changes following sleep deprivation. Scatter plots show the relationship between right postcentral and right inferior temporal functional connectivity changes and risk value changes following sleep deprivation. Corresponding functional connectivity is illustrated to the left, beside each scatter plot, in which blue represents right postcentral (69, −6, 21), red represents right inferior temporal (48, −18, −21) and green represents left middle temporal (−51, 0, −15). SD, sleep deprivation; RW, rested wakefulness; R, right; Inf, inferior.

**Table 1 brainsci-13-00514-t001:** Basic statistics of risky decision performance in both RW and sleep deprivation.

Variable, Mean (SD)	RW	Sleep Deprivation	*t*	*p*
Number of explosions	12.53 (3.86)	11.68 (3.95)	−1.13	0.27
BART values	12.12 (2.31)	11.80 (2.43)	−0.69	0.50
Total earnings	65.04 (4.96)	63.78 (5.68)	−0.87	0.40
Risk values	0.61 (0.07)	0.58 (0.08)	−2.19	0.04

Abbreviations: RW, rested wakefulness; BART, balloon analog risk task; SD, standard deviation.

**Table 2 brainsci-13-00514-t002:** Altered regional homogeneity following sleep deprivation.

	MNI Coordinates				Cluster Size	Brodmann Area
Regions	X	Y	Z	*t*		
Increased ReHo following SD						
Postcentral_R	69	−6	21	7.03 *	103	42

Abbreviations: SD, sleep deprivation; Reho, regional homogeneity; R, right; p-FDR, false discovery rate. * p-FDR < 0.05.

**Table 3 brainsci-13-00514-t003:** Altered right Postcentral functional connectivity following sleep deprivation.

	MNI Coordinates				Cluster Size	Brodmann Area
Regions	X	Y	Z	*t*		
Seed region Postcentral_R	69	−6	21			
Temporal_Mid_L	−51	0	−15	7.04 *	69	21
Temporal_Inf_R	48	−18	−21	6.94 *	100	21

Abbreviations: R, right; L, left; Mid, middle; Inf, inferior; p-FDR, false discovery rate. * p-FDR < 0.05.

## Data Availability

The datasets generated for this study are available on request to corresponding authors.

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
