# Peer review of "Altered Postcentral Connectivity after Sleep Deprivation Correlates to Impaired Risk Perception: A Resting-State Functional Magnetic Resonance Imaging Study"

_brainsci, 2023, doi:10.3390/brainsci13030514_

Round 1
Reviewer 1 Report
TITLE: Altered Postcentral Connectivity after Sleep Deprivation Correlates to Impaired Risk Perception: A Resting-State Functional Magnetic Resonance Imaging Study
The manuscript aimed to investigate the neuropsychological mechanisms of Sleep Deprivation-impaired risk perception from the perspectives of functional specialization and integration.
General concept comments
The manuscript is novel and unique and would be of interest to the readers of Brian Sciences.
First of all, a native or highly fluent English writer should assist with grammatical issues.
Abstract
Please in the summary delete the abbreviations, as SD, indicate the entire term.
Sleep deprivation (SD)
“….. of functional specialization and integration;
delete the semi-colon after integration and change with dot
the same thing in the entire Abstract
Introduction
First, it is essential to advance the argument/justification about the need for conducting this study. The claim about "the lack of studies" neither suffices (as lack of studies is not a strong argument in itself), nor seems to be factual given the amount of literature on the topic.
The objective is extensive and very wordy. It is also written differently throughout the manuscript (see abstract for instance). Please be consistent.
Methods
The note " The Ethics Committee of Beijing Sport University (Beijing, China) approved this study" is very general so please include the "protocol number" attesting that the study was approved by the Institutional Review Board / Human Subjects Committee. Participants permissions?
Did a medical doctor examine the participant? If yes, was always the same doctor?? Health record?
Results
A zero should not be inserted before a decimal fraction when the number cannot be greater than 1. For example, p < 0.05 should be written as “p < .05.” Continues in the same way!
Typically, if the exact p value is less than .001, you can merely state p < .001.
The conclusions are consistent with the evidence and arguments presented. Moreover, the author addresses the central questions posed.
Author Response
Please see the attachment. Thank you for your sincere suggestions.

Reviewer 2 Report
Thank you for recommending me as a reviewer. In this paper, the authors investigated the neuropsychological mechanisms of SD injury risk perception from the perspective of functional specialization and consolidation. Nineteen healthy male adults were recruited to undergo resting functional magnetic resonance imaging at rest and after a total SD of nearly 36 h at rest. In this paper, the authors found that SD impairs risk perception associated with altered postcentral connectivity. If the authors complete minor revisions, the quality of the study will be further improved.
1. The introduction section is well written. But that's too verbose. If the authors revise the introduction section to be more concise, it can help readers understand.
2. line 142: Where possible, authors should be more specific about the subject.
3. The authors described the limitations of the study well in the discussion section.
Author Response
Please see the attachment. Thank you for your sincere suggestion,
